# KIND: Blending Stationary and Transient Koopman Dynamics with Learned Uncertainty

## Abstract

We introduce KIND, a hybrid forecasting architecture that produces uncertainty-aware predictions by blending stationary and transient dynamics through Koopman operators. Classical system identification relies on fixed models of predictable behavior, while modern neural forecasting often sacrifices structure for flexibility. KIND operates at their intersection: it decomposes time series into components governed by stationary (model-driven) and transient (attention-driven) dynamics. These components are modeled independently via Koopman operators acting on lifted embeddings and fused through a Kalman-inspired, uncertainty-weighted blending mechanism. Crucially, KIND learns to estimate its own predictive confidence, enabling it to identify unfamiliar dynamics, adapt its forecasts, and express when its outputs can be trusted. A physics-informed pretraining strategy further strengthens its stationary pathway, encouraging robust separation of dynamic modes. Across both real-world and benchmark datasets—including superconducting radio frequency cavity measurements and electricity load forecasting—KIND demonstrates competitive accuracy, interpretable uncertainty, and resilience to distributional shifts, outperforming classical methods like Online DMD and competing neural models such as Koopa.

## 1 Introduction

Learning reliable models from time series data remains a central challenge in machine learning, particularly in the presence of distribution shifts—i.e., when the underlying dynamics evolve over time or change abruptly. Such shifts are common in real-world settings, from physical systems to financial markets, yet most modeling approaches assume fixed dynamics. This mismatch between training and deployment can degrade performance and erode model trustworthiness.

Recent work in operator-theoretic learning and neural forecasting seeks to bridge this gap by combining data-driven models with structured dynamics—for instance, Koopa employs hierarchical frequency decomposition and Koopman predictors (Liu et al., 2023), and SKOLR identifies structured Koopman dynamics with linear RNN architectures using spectral measurement functions (Zhang et al., 2025). Yet despite their architectural sophistication, most approaches remain limited to point estimates, lacking the ability to quantify uncertainty in their predictions. This limits their applicability in safety-critical and physically grounded domains, where understanding *when* and *why* a model is uncertain is as important as the forecast itself. Our work targets this gap.

One compelling example arises in the modeling of superconducting radio frequency (SRF) cavities, critical components in modern particle accelerators. In these systems, even minor environmental perturbations can shift the cavity's resonance frequency, a phenomenon known as detuning (Neumann et al., 2010). While detuning is typically represented as a scalar time series, it reflects a complex, high-dimensional interplay of mechanical vibrations, thermal fluctuations, and electromagnetic interactions. This creates a many-to-one mapping from the latent system state to a single observable, making the modeling problem deceptively difficult. Compounding the challenge, detuning is not directly measured, but instead inferred from multiple RF waveforms (e.g., forward and transmitted voltages). Thus, the apparent simplicity of the 1D signal belies a rich underlying dynamical structure—making it a natural testbed for interpretable and adaptive modeling techniques.

Classical modeling approaches, whether physics-based or data-driven, often fail to capture transition behaviors such as thermal drift or mechanical shocks. Operator-based learning methods like

Dynamic Mode Decomposition (DMD) (Wang, 2023) or Koopman neural models (Lusch et al., 2018) are well-suited to modeling stationary dynamics, but struggle with non-stationary or abrupt changes. Conversely, deep sequence models like Transformers (Nie et al., 2023) adapt well to shifting regimes, but often ignore structural priors and lack uncertainty quantification. This motivates a hybrid approach—one that can separate interpretable global dynamics from flexible, transient behavior, while also quantifying confidence in its outputs.

To address these challenges, we propose Kalman-Inspired Neural Decomposition (KIND)—a hybrid architecture that blends a global Koopman operator with a local Transformer-based operator via a dynamic weighting scheme inspired by Kalman filtering. The global operator captures stationary dynamics that represent prior beliefs about a system—a model—but since no model is perfect, the local operator adapts in real-time to distribution shifts or regime changes. These two predictions are fused through a confidence-aware blending mechanism, using learned uncertainty to determine which operator to trust at each time step. The result is a system that is interpretable, adaptive, and uncertainty-aware.

**Our contributions are:**

- We introduce KIND, a neural architecture that explicitly separates and blends stationary and transient dynamics using operator-theoretic structure.

- We propose a principled uncertainty modeling mechanism for each branch, trained using a curriculum-based approach to estimate epistemic confidence per forecast slice.

- We show that KIND achieves interpretable uncertainty-aware forecasting on both real-world and benchmark datasets, and supports direct comparisons with classical models such as DMD and modern neural baselines like Koopa.

- We demonstrate the method on real-world SRF cavity data from a particle accelerator, showcasing its applicability to physical systems with latent complexity.

## 2 RELATED WORK

**Koopman-based Kalman filtering.** Koopman-based state estimation has been applied to particle accelerator systems. Syed et al. (2021) proposed a fault detection scheme for SRF cavities using Extended DMD and physics-informed basis functions to build a linear model for a Kalman filter. Faults were detected via a chi-squared test on the innovation signal. While effective, this method assumes fixed thresholds and Gaussian noise. In contrast, our framework blends global Koopman structure with locally adaptive dynamics, and supports uncertainty-aware anomaly detection without explicit statistical tests.

**DMD-based cavity control.** Wang (2023) applied kernel DMD with polynomial features for SRF cavity stabilization under a model predictive control scheme. Their approach captures moderate nonlinearities using a fixed dictionary and assumes stationary dynamics. KIND differs by explicitly modeling transients via adaptive operators, making it more suitable under distribution shift and potentially compatible as a plug-in forecasting layer for downstream control.

**Adaptive Transformers and operator fusion.** Wang et al. (2023) introduced a Transformer-based Koopman model that learns local linearizations for time-varying dynamics. Our work extends this idea through a Kalman-inspired fusion of stationary and adaptive pathways, with learned uncertainty guiding their combination. Unlike their additive design, we impose an interpretable weighting mechanism that reflects confidence in each component.

## 3 BACKGROUND

### 3.1 TIME SERIES FORECASTING VIA KOOPMAN THEORY

Time series data $\{x_t\}_{t=1}^T$ can be treated as observables of a state in a dynamical system. These observations evolve according to a discrete-time process

$$x_{t+1} = f(x_t), \quad x_t \in \mathcal{X} \subset \mathbb{R}^d, \tag{1}$$

where $x_t$ is the state at time $t$, and where $\mathcal{X}$ is the set of all physically realizable real-valued states. The model of system dynamics is denoted with $f$. The direct modeling of $f$ is often intractable due to nonlinearities present in real-world systems. Koopman operator theory (Koopman, 1931) offers an alternative: instead of modeling the nonlinear dynamics in the original state space $\mathcal{X}$, the system is lifted into a higher-dimensional observable space, where the dynamics evolve linearly. Formally, there exists a linear infinite-dimensional operator $\mathcal{K}$ that acts on a set of observable functions $g : \mathcal{X} \to \mathbb{R}$ to advance these observations in time, i.e.,

$$\mathcal{K}g(x_t) = g(x_{t+1}) = g(f(x_t)). \tag{2}$$

## 3.2 DYNAMIC MODE DECOMPOSITION

Although the Koopman operator $\mathcal{K}$ provides a powerful linear representation of nonlinear dynamics, it is infinite-dimensional in general and thus cannot be computed directly. To overcome this, DMD (Schmid, 2010) offers a practical, data-driven approach to approximate a finite-dimensional version of $\mathcal{K}$.

Let $\mathbf{Z} = [g(x_1), g(x_2), \ldots, g(x_{T-1})]$ and $\mathbf{Z}' = [g(x_2), g(x_3), \ldots, g(x_T)]$. The goal of DMD is to find a linear operator $\mathcal{K}$ such that

$$\mathbf{Z}' \approx \mathcal{K}\mathbf{Z}, \tag{3}$$

which leads to the least-squares solution

$$\mathcal{K} = \mathbf{Z}'\mathbf{Z}^\dagger, \tag{4}$$

where $\mathbf{Z}^\dagger$ denotes the (pseudo)inverse of $\mathbf{Z}$.

Modern extensions, such as the deep Koopman framework by Lusch et al. (2018), replace explicit inversion with neural networks that learn both the lifting function $g(x)$ and the linear operator $\mathcal{K}$, enabling more flexible and expressive representations in high-dimensional or partially observed systems.

## 3.3 TRANSFORMERS FOR TIME-VARYING DYNAMICS

Despite their utility as practical Koopman approximations, DMD-based solutions have a core limitation: they assume stationary system dynamics. In real-world settings, local, time-varying phenomena—such as regime shifts, environmental perturbations, or transient faults—can render a fixed linear operator $\mathcal{K}$ inaccurate or unstable for forecasting. To address this, Transformer architectures have recently demonstrated strong empirical performance in time series forecasting under distribution shifts.

While Transformers are often used as direct sequence-to-sequence forecasting models, a more structured and interpretable approach is to use them as adaptive operator generators. Specifically, instead of predicting future states outright, a Transformer can be employed to produce a sequence of time-varying evolution matrices that propagate a system state—or its lifted representation—forward in time, e.g.

$$\mathcal{K}_t = \gamma(g(x_t)), \quad \mathcal{K}_t \in \mathbb{R}^{n \times n}, \tag{5}$$

where $g(x_t)$ is a lifted representation of the input, $n$ denotes lifted dimensionality and $\gamma$ comprises a compact Transformer encoder followed by a small MLP. This formulation maintains a Koopman-inspired linear evolution structure while allowing the dynamics to evolve flexibly across time.

When such an adaptive Transformer-based operator is combined with a stationary DMD-like baseline, the result is a hybrid model architecture that balances global consistency with local adaptability—a core principle behind our proposed KIND framework.

# 4 KIND

## 4.1 FROM STRUCTURE TO STRATEGY

Classical system identification assumes a fixed and often simplified model of system dynamics, while modern neural forecasting often treats structure as a liability, opting instead for end-to-end learning. KIND takes a middle ground: we decompose the observed time series into two interacting components—stationary and transient—each modeled by a dedicated architectural branch with its own learning dynamics.

This split is central to both KIND's design and its training strategy. The stationary branch is tasked with capturing persistent dynamics that align with domain knowledge or spectral priors. In contrast, the transient branch is responsible for modeling non-stationary fluctuations, abrupt regime changes, or rare events.

To facilitate this separation, we follow a staged training procedure:

1. **Simulated stationary dynamics.** We first construct synthetic data based on known physics or frequency content analysis. This data captures the modeled, stationary component of the dynamics.

2. **Stationary pretraining.** We train only the stationary branch of KIND on this simulated data, keeping the transient branch frozen. We record the maximum mean squared error (MSE) across this training set to use as a filtering threshold.

3. **Data filtering.** Real measurement data is passed through the trained stationary model. Input windows are labeled as stationary if their reconstruction error is below the threshold, or as transient otherwise.

4. **Final training.** Using this labeled dataset, we train both branches of KIND jointly on their corresponding data types.

Once trained, each branch of KIND operates on its own latent representation of the input. These representations evolve independently under separate Koopman dynamics and are fused into a final prediction via a learned uncertainty-aware blending mechanism. We now describe the architecture of these modeling branches and their interactions.

## 4.2 STATIONARY AND TRANSIENT KOOPMAN OPERATORS WITH BLENDING

KIND maintains two complementary modeling pathways: the stationary branch, and the transient branch. Each branch operates over its own latent representation of the input, denoted $\Xi^{\text{stat}}$ and $\Xi^{\text{trans}}$, respectively. These embeddings are computed from raw time series via a lifting process based on predefined basis functions and slice-wise encoders. The lifting process is detailed in the next subsection, here we just note that each embedding $\Xi$ consists of a sequence of lifted slices $\Xi_j$, where $j$ indexes non-overlapping temporal windows of the input data.

We model each embedding's evolution with a Koopman operator: a fixed linear operator $\mathcal{K}^{\text{stat}}$ for the stationary branch, and a dynamically inferred operator $\mathcal{K}^{\text{trans}}_j$ for the transient branch, predicted via a network $\gamma$. As noted in Section 3.3, $\gamma$ produces a sequence of operator matrices, hence the slice index $j$ in $\mathcal{K}^{\text{trans}}_j$. Finally, each evolved embedding is decoded back into the time series space via respective decoders $\psi^{\text{stat}}$ and $\psi^{\text{trans}}$.

To fuse these outputs, we introduce a learned uncertainty estimate per branch, $\zeta^{\text{stat}}_{j+1}$ and $\zeta^{\text{trans}}_{j+1}$, trained via Gaussian negative log-likelihood. These uncertainties act on a slice level and control a blending coefficient $\alpha_{j+1} \in [0, 1]$ defined as

$$\alpha_{j+1} = \frac{\zeta^{\text{trans}}_{j+1}}{\zeta^{\text{stat}}_{j+1} + \zeta^{\text{trans}}_{j+1}}. \tag{6}$$

We use the slice index $j+1$, because $\zeta_{j+1}$ estimates the uncertainty of a predicted $\Xi_{j+1}$ embedding. The final prediction is then given by

$$\hat{X}_{j+1} = \alpha_{j+1} \cdot \hat{X}_{j+1}^{\text{stat}} + (1 - \alpha_{j+1}) \cdot \hat{X}_{j+1}^{\text{trans}}, \tag{7}$$

where each $\hat{X}_{j+1}$ is obtained by evolving and decoding the embedding

$$\hat{X}_{j+1}^{\text{stat}} = \psi^{\text{stat}}(\mathcal{K}^{\text{stat}} \Xi_j^{\text{stat}}), \qquad \hat{X}_{j+1}^{\text{trans}} = \psi^{\text{trans}}(\mathcal{K}_j^{\text{trans}} \Xi_j^{\text{trans}}). \tag{8}$$

This uncertainty-based blending is reminiscent of a Kalman update: the stationary model acts as a prior, while the transient branch plays the role of a measurement. When the transient uncertainty is low, its prediction dominates; when the stationary model is more confident, the system falls back to stable dynamics. Unlike classical filters, however, KIND learns these uncertainties from data, enabling heteroscedastic and context-aware updates. See Figure 1 for an overview of the architecture.

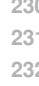
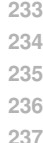


Figure 1: KIND architecture. Block diagram illustrating the structure of KIND, which fuses two forecasting branches: a stationary DMD-like model (top section) and a transient Transformer-based model (bottom section). The input time series slice $X_j$ is encoded independently by each branch into latent states $\Xi_j^{\text{stat}}$ and $\Xi_j^{\text{trans}}$. These are then advanced to produce $\Xi_{j+1}^{\text{stat}}$ and $\Xi_{j+1}^{\text{trans}}$, decoded into forecasts, and blended using uncertainty-weighted fusion to yield the final prediction $\hat{X}_{j+1}$. While the diagram shows a single slice $j$ for clarity, the model operates over all available slices during training and inference.

Central to both branches is the lifting process that transforms raw time series data into Koopman-compatible latent embeddings. We now describe this transformation in more detail.

### 4.3 LIFTED REPRESENTATIONS VIA BASIS FUNCTIONS

Each branch in KIND begins by transforming raw time series data into lifted latent embeddings $\Xi^{\text{stat}}$ and $\Xi^{\text{trans}}$. This is accomplished through a basis-function-driven lifting process designed to approximate Koopman-invariant subspaces.

We define a function family $\mathcal{G} := \{g_i\}_{i=1}^n$, which can be either shared across branches or be branch-specific. To extract parameters for these basis functions, we segment the time series $\{x_t\}_{t=1}^T$ into $m$ non-overlapping $\tau$-sized slices $\{X_j\}_{j=1}^m \subset \mathbb{R}^{\tau \times d}$, where each slice $X_j$ provides localized temporal context. The slice length $\tau$ governs a trade-off: longer slices capture global structure but may miss transients; shorter slices improve responsiveness.

A branch-specific kernel encoder $\varphi$ maps each slice $X_j$ to a latent representation that modulates the basis function parameters, yielding $V \in \mathbb{R}^{n \times m}$. For instance, a parameter for the $i$-th function in the $j$-th slice is computed as

$$v_{i,j} = \varphi(X_j)^T X_j. \tag{9}$$

Once parameterized, each basis function $g_i$ is evaluated to produce latent embeddings $\xi_{i,j} = g_i(v_{i,j})$, yielding a lifted matrix $\Xi \in \mathbb{R}^{n \times m}$. This process of lifting time series data into latent embeddings is illustrated in Figure 2.

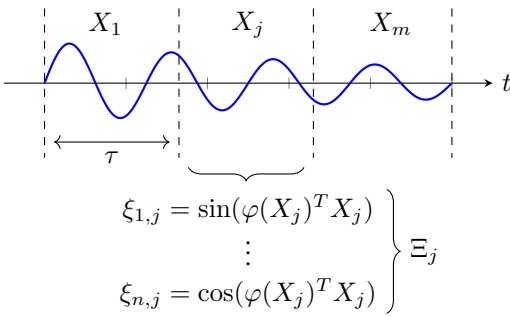

Figure 2: Lifting of time series data into latent embeddings. The data is split into $m$ $\tau$-sized slices, and each slice is subjected to $n$ basis functions, e.g., a $\sin$ and a $\cos$, to form a latent embedding $\Xi$.

Temporal segmentation via fixed-size slices plays a dual role in KIND. First, it ensures that each lifted embedding captures a temporally coherent segment of the signal, suitable for structured basis function parameterization. Second, it enables computational efficiency by compressing long input sequences into a smaller number of latent steps, reducing the number of Koopman operator evaluations needed for prediction. As a result, this design balances modeling fidelity with tractability, and it permits branch-specific slice lengths—offering coarse global embeddings in the stationary path and finer local responsiveness in the transient path.

While each branch provides a standalone forecast, their predictions are ultimately fused based on learned confidence estimates. These uncertainties not only guide blending but also serve as indicators of each model's reliability in varying regimes. We now describe how uncertainty is modeled and trained within KIND.

### 4.4 UNCERTAINTY MODELING VIA BRANCH-SPECIFIC VARIANCE MODULES

A key design goal in KIND is to quantify epistemic uncertainty in a principled and interpretable manner. This is achieved through two learned uncertainty modules—$\zeta^{\text{stat}}$ and $\zeta^{\text{trans}}$—which estimate the predictive variance of their respective Koopman branches. These modules produce uncertainty estimates that drive the blending weight $\alpha$ in (6) via a heteroscedastic Gaussian likelihood.

Training these modules proceeds in four stages, each targeting a different aspect of the model:

1. Stationary mean training. The stationary Koopman operator $\mathcal{K}^{\text{stat}}$ is trained on data exhibiting globally persistent dynamics, using mean squared error (MSE) loss.

2. Stationary variance training. With $\mathcal{K}^{\text{stat}}$ frozen, the variance module $\zeta^{\text{stat}}$ is trained using negative log-likelihood (NLL) loss on mixed data that includes transient perturbations. This stage teaches the stationary branch to detect and quantify its own inadequacy in nonstationary regimes.

3. Transient mean training. The transient Koopman operator $\mathcal{K}^{\text{trans}}$ is trained on data with high-frequency or transient content, again using MSE loss.

4. Transient variance training. With $\mathcal{K}^{\text{trans}}$ frozen, the transient variance module $\zeta^{\text{trans}}$ is trained—again using NLL—on the same mixed data. This captures how well the transient branch generalizes outside its training regime.

This training curriculum explicitly teaches each branch what it does not know, enabling more meaningful uncertainty calibration during inference.

## 5 EXPERIMENTS

**Datasets.** We evaluate KIND on two datasets: ETTh2 (Zhou et al., 2021) and a TESLA cavity dataset. ETTh2 is a public benchmark dataset for long-term time series forecasting based on hourly electricity transformer data, where we focus on univariate prediction of oil temperature. In contrast, the TESLA dataset involves cavity detuning measurements, offering a scientific signal with a mix of harmonic and transient structure.

We focus on univariate forecasting (oil temperature) in ETTh2 to better isolate and interpret the behavior of the stationary component, which is sensitive to dominant periodic dynamics. Extending to multivariate settings is left for future work. We also include an evaluation on the M4 Hourly dataset (Makridakis et al., 2018) to test KIND in heterogeneous ensemble settings; see Appendix B.

**Baselines.** We compare KIND against two established baselines. First, Online DMD (Zhang et al., 2019) is a purely dynamic model that updates its linear operator in an online fashion using recursive least squares (RLS). While designed for adaptive identification, it lacks forecasting capability once new measurements cease. Second, Koopa (Liu et al., 2023) is a frequency-aware hybrid model that leverages Fourier filtering and neural operators to handle distribution shifts.

To further understand KIND's performance, we also evaluate its internal components independently. The stationary branch resembles classical DMD, trained on known dynamics, while the transient branch is a Transformer that adapts to novel patterns. These variants serve as ablation baselines and allow us to assess the impact of uncertainty-guided blending.

**Setup.** We evaluate KIND and all baselines using a rolling window strategy. For each dataset, we set a lookback window length $T$ and forecast horizon $H$ to maintain $T = 2H$, following common practice (Liu et al., 2023). Specifically, we use $T = 96$, $H = 48$ for ETTh2 and M4 Hourly, but $T = 80$, $H = 40$ for TESLA. Furthermore, for ETTh2, we perform a roughly 50/50 train-test split, with additional test data partitioning to isolate segments of falling and rising trends. TESLA is split in a 75/25 ratio without further sub-regimes.

We apply window-wise min-max scaling to map each input time series window to a range $[-1, 1]$. This scaling preserves local variation, including slope and trend information, and enables unnormalization after prediction to recover physical units. Unlike some baselines (e.g., Koopa), we do not apply global or double normalization. Finally, all results are averaged across 3 independent runs with different seeds. Implementation details are provided in Appendix A.

### 5.1 TIME SERIES FORECASTING

**Online DMD: divergence under open-loop rollout.** Online DMD (Zhang et al., 2019) is a recursive method designed for continual system identification. It relies on recursive least squares (RLS) to update its linear dynamics online, making it suitable for tracking evolving systems when continual measurements are available. However, as shown in Figure 3, Online DMD quickly diverges under open-loop forecasting, where no new observations are provided beyond the lookback window.

**KIND vs. Koopa: uncertainty-aware forecast evaluation.** We compare KIND and Koopa on the ETTh2 dataset (Zhou et al., 2021), evaluating both forecast accuracy and KIND's learned uncertainty estimates. To highlight model behavior across different regimes, we partition the test set into two segments: one with only falling trends and another with a mix of falling and rising slopes. Forecasting is done in *real physical units* (degrees Celsius) to ensure meaningful interpretation of both errors and uncertainties.

Table 1 reports results for this univariate setup, where the goal is to predict oil temperature from its historical values. This signal is especially suitable for Koopman-based modeling due to its periodic but nonstationary structure. Notably, KIND outperforms Koopa across both segments in terms of MSE and MAE, despite having the same number of parameters (~0.27M). This suggests that the additional uncertainty-aware mechanism does not come at the cost of predictive performance.

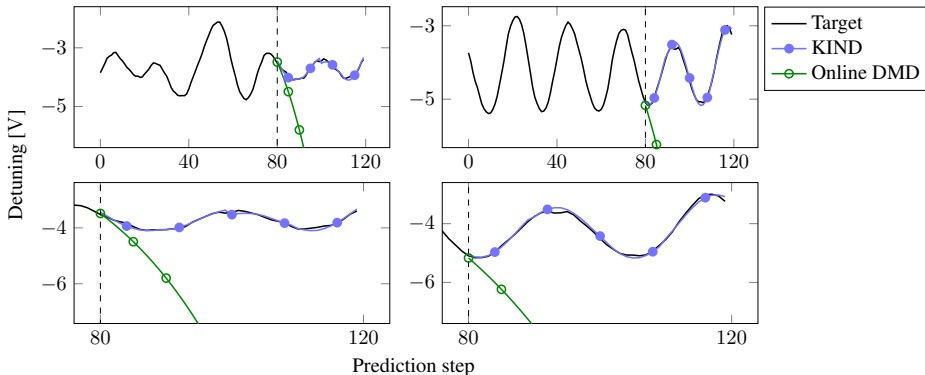

Figure 3: Example forecasts on the TESLA dataset (left and right). Online DMD diverges quickly when not continuously updated with new data. In contrast, KIND maintains trajectory fidelity across the entire forecast horizon.

Interestingly, the average uncertainty $\overline{\zeta^{\text{stat}}}$ assigned to the stationary branch increases from 0.330 (falling) to 0.353 (mixed), reflecting the greater modeling difficulty posed by rising slopes. This aligns with the relative increase in Koopa's error on the mixed regime, suggesting that KIND's uncertainty signals may also serve as useful indicators of model trustworthiness. Meanwhile, the transient branch maintains consistent uncertainty levels ($\overline{\zeta^{\text{trans}}} \approx 0.030$–$0.031$), underscoring its robustness to directional shifts.

Table 1: Univariate oil temperature forecasting on ETTh2, evaluated in real-world units (degrees Celsius). KIND slightly outperforms Koopa in both MSE and MAE across falling and mixed slope regimes, while also providing time-resolved uncertainty estimates $\overline{\zeta^{\text{stat}}}$ and $\overline{\zeta^{\text{trans}}}$. Higher uncertainty in the stationary component correlates with increased difficulty on rising trends.

| | Koopa | | KIND | | | |
|---|---|---|---|---|---|---|
| Data segment | MSE | MAE | MSE | MAE | $\overline{\zeta^{\text{stat}}}$ | $\overline{\zeta^{\text{trans}}}$ |
| Falling slope | 13.981 | 2.814 | 12.938 | 2.721 | 0.330 | 0.031 |
| Falling and rising slopes | 19.696 | 3.269 | 18.530 | 3.231 | 0.353 | 0.030 |

## 5.2 Ablation Study

**Component-level performance.** To isolate the contribution of each submodel in KIND's architecture, we evaluate the stationary branch, the transient branch, and the full uncertainty-weighted blend on both datasets. Table 2 reports normalized MSE and MAE for each configuration.

We observe that the full KIND model consistently outperforms its individual branches, highlighting the benefit of dynamic uncertainty-weighted blending. While the transient branch is already strong on its own, especially on the TESLA dataset, blending slightly improves performance in both datasets. Note however that KIND assumes a degree of temporal and structural homogeneity in the dataset; when applied to highly heterogeneous time series ensembles, the blending performance may degrade (see Appendix B).

**Blending effectiveness.** To visualize how KIND fuses stationary and transient forecasts, Figure 4 presents two example sequences from the TESLA dataset: one where both branches produce accurate predictions, and one where the stationary model fails. In both cases, the model's uncertainty estimates reflect the prediction quality of each branch: $\zeta^{\text{stat}}$ rises when the stationary forecast deviates, and $\zeta^{\text{trans}}$ remains low when the transient branch successfully adapts. The resulting blend weight $\alpha$ shifts accordingly. This illustrates that KIND does not simply average predictions—it selectively trusts the branch with lower epistemic uncertainty. Blending results on the ETTh2 dataset can be found in Appendix B.

Table 2: Normalized forecasting errors (MSE and MAE) for KIND and its standalone components. KIND consistently improves over its parts via uncertainty-aware blending.

| | ETTh2 (mixed slopes) | | TESLA | |
| Model | MSE | MAE | MSE | MAE |
|---|---|---|---|---|
| KIND (stationary only) | 0.296 | 0.424 | 0.214 | 0.349 |
| KIND (transient only) | 0.170 | 0.304 | 0.086 | 0.216 |
| KIND (blend) | **0.163** | **0.300** | **0.084** | **0.216** |

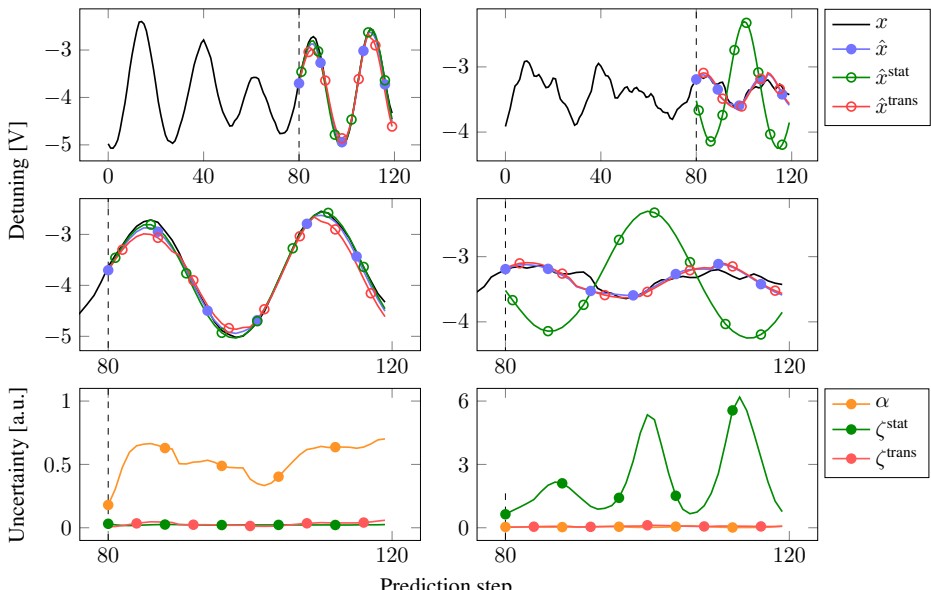

Figure 4: Examples of blending behavior on the TESLA dataset. When both branches agree (left), then $\alpha \approx 0.5$ reflects equal confidence. In case the stationary model diverges (right), $\zeta^{\text{stat}}$ increases, and $\alpha \to 0$, the transient forecast is thus favored.

## 6 CONCLUSION

In this work, we proposed KIND: a hybrid forecasting architecture that blends stationary and transient dynamics via uncertainty-weighted Koopman operators. A distinguishing feature of KIND is its ability to produce interpretable, time-resolved uncertainty estimates, which are often lacking in current operator-theoretic and neural forecasting models. While this capability requires additional steps—such as data simulation, stationarity filtering, and staged training—the resulting interpretability and adaptability offer significant advantages in task-driven applications, such as control and online monitoring.

We benchmarked KIND against both Online DMD, which diverges without continuous updates, and Koopa, a recent Koopman-based deep model. In real-unit evaluations, KIND slightly outperforms Koopa across different slope regimes, while also offering principled uncertainty modeling. Importantly, both models use a comparable number of parameters, making KIND a competitive and interpretable alternative for nonstationary time series forecasting.

Looking ahead, we plan to integrate KIND into model-based control frameworks, including Model Predictive Control (MPC) and Reinforcement Learning (RL), where uncertainty estimates can be directly exploited for planning and safety. Another promising direction is to explore theoretical stability guarantees, drawing inspiration from existing work on safe learning with Gaussian processes (Berkenkamp et al., 2017) and extending it to the structured operator learning setting.

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

## A   IMPLEMENTATION DETAILS

KIND is trained with L2 loss and optimized by ADAM (Kingma & Ba, 2015) with a learning rate of $10^{-3}$ and a weight decay of $10^{-4}$. Batch size is set to 128. The model is implemented using PyTorch (Paszke et al., 2019) and trained on an Apple M3 Max CPU.

## B   Supplementary experimental results

**Blending effectiveness on the ETTh2 dataset.**   Figure 5 visualizes blending and uncertainty behavior on the ETTh2 dataset. The same trends observed in TESLA apply: when the stationary model becomes inaccurate, $\zeta^{\text{stat}}$ increases and $\alpha$ shifts toward the transient forecast.

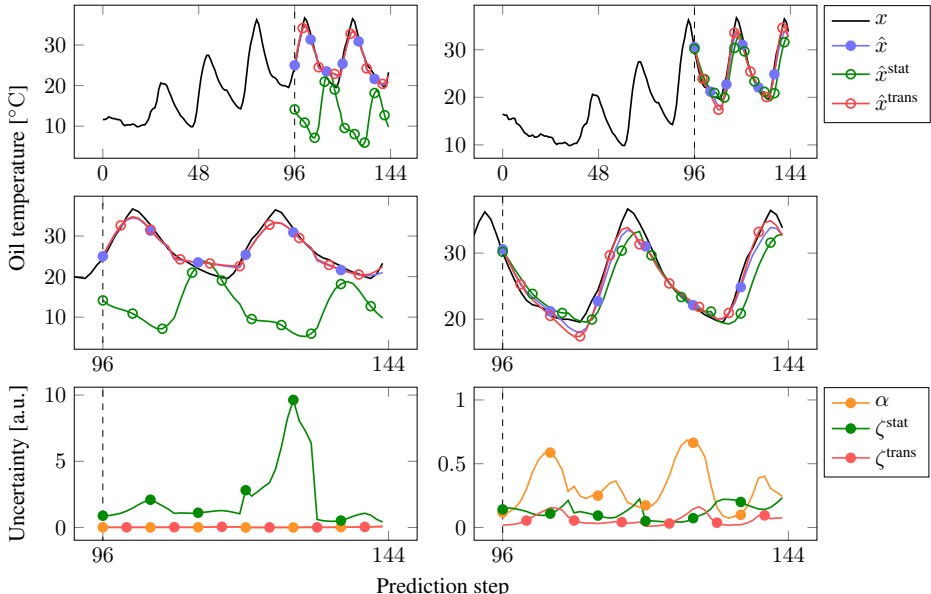

Figure 5: Examples of blending and uncertainty behavior on the ETTh2 dataset.

**Sensitivity to basis function choice.**   We examine how KIND's performance changes with different basis function types used in the stationary and transient branches. Specifically, we test combinations of data-driven and sinusoidal (Fourier) bases on the ETTh2 dataset. Table 3 reports the normalized MSE for each configuration.

Results indicate that the Transformer-based transient branch is robust to basis function variation, while the DMD-based stationary branch exhibits more variability. However, the full KIND model consistently maintains stable performance across all combinations, suggesting that the blend mechanism compensates for potential degradation in the stationary branch.

Table 3: Normalized MSE on ETTh2 for different basis function combinations in KIND. The transient branch remains robust across basis choices, and the blended model shows consistent performance.

| Basis | | MSE | | |
|---|---|---|---|---|
| Stationary | Transient | Blend | Stationary-only | Transient-only |
| data-driven | data-driven | 0.162 | 0.319 | 0.172 |
| data-driven | sinusoidal | 0.166 | 0.307 | 0.178 |
| sinusoidal | data-driven | 0.165 | 0.338 | 0.172 |
| sinusoidal | sinusoidal | 0.167 | 0.361 | 0.175 |

These results support the view that KIND's blending mechanism acts as a soft selector, relying more on the robust transient branch when stationary modeling is less reliable.

**Sensitivity to $\tau$ hyperparameter choice.**   As noted in Section 4.3, the hyperparameter $\tau$ controls the temporal resolution perceived by Koopman operators in both branches. Intuitively, a larger $\tau$ should help the stationary branch capture slower, stationary dynamics by allowing the operator to

observe coarser transitions. However, as shown in Figure 6, on the ETTh2 dataset, smaller values of $\tau$ result in better overall performance for the full blended KIND model.

We hypothesize that this behavior stems from the uncertainty estimation mechanism. A smaller $\tau$ leads to sharper and more responsive $\zeta^{\text{stat}}$ signals, which improves the calibration of the blend weight $\alpha$. This allows the model to downweight poor stationary forecasts more quickly and trust the transient component when needed. Hence, even if the stationary operator is not significantly improved, the blending becomes more accurate.

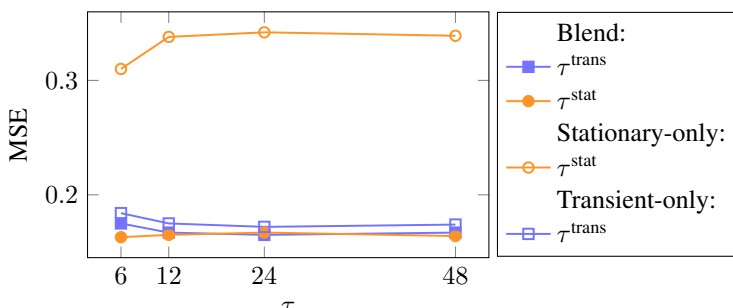

Figure 6: Normalized MSE on the ETTh2 dataset for different values of $\tau$ used in the stationary and transient operators. During the sweep of $\tau^{\text{stat}}$, $\tau^{\text{trans}}$ was held fixed, and vice versa.

**Limitations of KIND in heterogeneous time series ensembles.** The current version of KIND is designed for task-specific scenarios where a single dominant stationary process (e.g., a physical system or control loop) is occasionally disrupted by transient deviations. This structure aligns with control-theoretic settings and is effective on datasets like TESLA and ETTh2, which share such dynamics.

However, in datasets such as M4 Hourly (Makridakis et al., 2018), composed of hundreds of unrelated univariate time series with diverse behaviors, this assumption no longer holds. A single shared Koopman operator cannot model all stationary regimes simultaneously, and the uncertainty-based blending mechanism—which acts as a soft selector—may not fully suppress inaccurate stationary predictions when uncertainty estimates fail to sharply discriminate.

As a result, the transient-only model typically outperforms the full blended architecture, as shown in Table 4. Figure 7 presents two representative examples where the stationary component slightly degrades the forecast, despite high uncertainty. While the degradation is not catastrophic, it highlights the current model's limitation in handling process heterogeneity via soft blending.

Table 4: Forecasting errors (sMAPE and MAE) for KIND and its components on the M4 Hourly dataset. The transient component performs best, indicating limited utility of the stationary model in this heterogeneous setting.

| Model | sMAPE | MAE |
|---|---|---|
| KIND (stationary only) | 26.647 | 0.120 |
| KIND (transient only) | **14.054** | **0.077** |
| KIND (blend) | 14.298 | 0.080 |

Future work could address this limitation by learning multiple stationary models through clustering, gating mechanisms, or mixture-of-experts architectures. This would allow KIND to transition from a task-specific design to an ensemble-aware formulation, similar to gain scheduling strategies in control theory.

## C    USE OF LARGE LANGUAGE MODELS (LLMs)

Large language models were used during the writing of this paper solely for language polishing, clarity improvements, and structuring suggestions. No generated text was used as scientific content

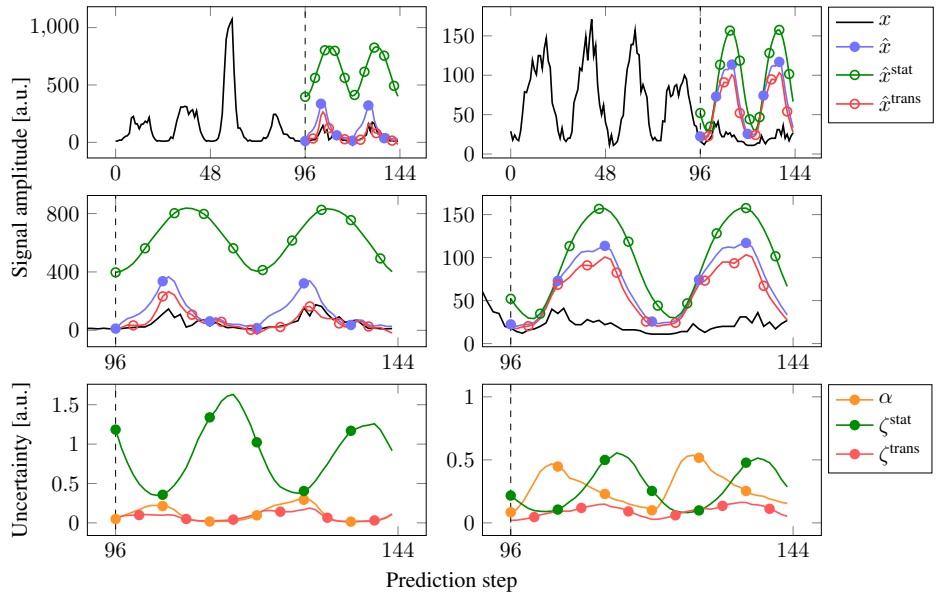

Figure 7: Examples from M4 Hourly where the stationary forecast degrades the blend. While $\alpha$ does downweight the stationary component, it remains too soft to fully suppress its influence.

or empirical evidence. All technical contributions, experimental designs, models, analyses, and conclusions are original and authored by the listed contributors.