# OpenReview forum: "KIND: Blending Stationary and Transient Koopman Dynamics with Learned Uncertainty"
_ICLR.cc/2026/Conference — ICLR 2026 Conference Withdrawn Submission_

### Official Review · Reviewer_bYUX · 2025-10-23

**Soundness:** 2
**Presentation:** 2
**Contribution:** 2
**Rating:** 2
**Confidence:** 4

**Summary:**

This paper proposes KIND (Kalman-Inspired Neural Decomposition), a hybrid forecasting framework that integrates Koopman operator theory with uncertainty-aware neural modeling. The key idea is to decompose time series dynamics into two complementary components:

a stationary branch, modeled via a fixed Koopman operator to capture stable, global dynamics; and

a transient branch, implemented as a Transformer-based adaptive operator that accounts for local or non-stationary changes.

The two branches are combined using a Kalman-inspired blending mechanism, where learned uncertainty estimates determine the confidence-weighted fusion of predictions. This approach allows the model to adapt to distribution shifts while maintaining interpretability and providing calibrated epistemic uncertainty.

To train the system, the authors employ a staged curriculum involving simulated stationary pretraining, stationarity-based data filtering, and joint optimization of both branches. The method is evaluated on ETTh2 (electricity transformer temperature), TESLA SRF cavity detuning, and M4 Hourly datasets. Experimental results show that KIND achieves competitive or superior performance to baselines such as Koopa and Online DMD, while offering meaningful uncertainty quantification and robustness under nonstationary regimes. Ablation studies confirm that the uncertainty-weighted blending improves performance over individual branches.

The paper also discusses limitations, noting that KIND assumes a dominant stationary process and may underperform in highly heterogeneous ensembles. Future work aims to extend the model toward mixture-of-experts structures and integration with model-based control frameworks (e.g., MPC, RL).

**Strengths:**

1. The paper introduces a novel hybrid forecasting framework, KIND, that explicitly blends stationary and transient dynamics through a Koopman-operator-based structure with learned uncertainty weighting. While Koopman models and Transformer forecasters have been studied independently, their integration via a Kalman-inspired uncertainty-driven fusion is both conceptually elegant and practically innovative. The idea of treating uncertainty as a control variable for model blending represents a creative extension of operator-theoretic learning.
2. The paper is clearly structured and easy to follow. Conceptual diagrams (e.g., Figure 1) effectively communicate the model architecture and blending process. The authors maintain a coherent narrative throughout, explaining each design choice with intuition grounded in system identification and control theory. The writing is precise and avoids unnecessary complexity despite the technical depth of the topic.
3. KIND addresses a practically important and underexplored gap in time-series forecasting—how to jointly model interpretable stationary dynamics and adaptive transient behaviors under uncertainty. The approach has strong implications for physics-informed machine learning, control systems, and nonstationary signal forecasting. By demonstrating competitive accuracy alongside interpretable uncertainty quantification, the paper advances the field toward more trustworthy and adaptive operator-theoretic models.

**Weaknesses:**

1. The experimental validation of the paper is somewhat limited. The authors compare KIND only against Koopa and Online DMD, which does not provide a sufficiently comprehensive evaluation. To convincingly demonstrate the model’s advantages, the paper should include comparisons with more recent Koopman-based approaches such as SKOLR [1] and Koopman Neural Forecaster (KNF) [3], as well as state-of-the-art time-series forecasting models from the broader literature, such as iTransformer [2]. Moreover, the experimental datasets—ETTh2, TESLA cavity, and M4 Hourly—are too few and domain-specific. Evaluations on a wider variety of datasets (e.g., weather, traffic, or financial time series) would make the empirical evidence more robust and generalizable.
2. Although the paper’s motivation and title emphasize forecasting under non-stationary dynamics, there is no quantitative measurement of non-stationarity to support the claims. Without such metrics, it is difficult to assess whether KIND indeed performs better on highly non-stationary data. Incorporating statistical indicators such as the Augmented Dickey–Fuller (ADF) test or related non-stationarity measures, and reporting performance across varying degrees of non-stationarity, would strengthen the empirical argument.
3. The related work section is not sufficiently comprehensive. The discussion omits several relevant advances in both general time-series forecasting (e.g., iTransformer [2]) and Koopman-based modeling (e.g., KNF [3], SKOLR [1]). A more systematic comparison to these recent methods would help clarify the novelty and positioning of KIND within the broader landscape of operator-theoretic and neural forecasting models.
4. The paper provides too few visual comparisons. The qualitative results primarily focus on Online DMD, while omitting visual comparisons with Koopa or other baselines. Including more visual case studies, especially in complex or transient regimes, would make the interpretability and uncertainty-awareness claims more convincing. Richer qualitative analysis—e.g., uncertainty evolution plots, regime transitions, and multi-model overlays—would substantially enhance the clarity and impact of the experimental section.

[1] SKOLR: Structured Koopman Operator Linear RNN for Time-Series Forecasting

[2] iTransformer: Inverted Transformers Are Effective for Time Series Forecasting

[3] Koopman Neural Forecaster for Time Series with Temporal Distribution Shifts

**Questions:**

N/A

---

### Official Review · Reviewer_xgBD · 2025-10-26

**Soundness:** 3
**Presentation:** 2
**Contribution:** 2
**Rating:** 4
**Confidence:** 4

**Summary:**

The authors propose KIND, a hybrid forecasting architecture that generates uncertainty-aware predictions by integrating stationary and transient dynamics through Koopman operators.  KIND demonstrates competitive accuracy, interpretable uncertainty, and resilience to distributional shifts, outperforming classical approaches like Online DMD and neural baselines such as Koopa.

**Strengths:**

1.KIND achieves interpretable uncertainty-aware forecasting on both realworld and benchmark datasets

**Weaknesses:**

1.The experiments and baselines are relatively limited; it is necessary to conduct experiments on more datasets and expand the range of compared baselines, e.g., [1] [2] [3] [4].

this paper is also based on koopman and kalman operators, while it also provides quantitive uncertainty and decomposing time series data into different branches.
[1] K2VAE: A Koopman-Kalman Enhanced Variational AutoEncoder for Probabilistic Time Series Forecasting, ICML 2025.

these papers are current sota tsf baslines
[2] Are Transformers Effective for Time Series Forecasting?, AAAI 2023
[3] A Time Series is Worth 64 Words: Long-term Forecasting with Transformers, ICLR 2023
[4] TimesNet: Temporal 2D-Variation Modeling for General Time Series Analysis, ICLR 2023

As for datasets, you should consider tsf datasets from different areas including ecl, exchange, weather, traffic and other ETTs.



2. The method of dividing the data into a stationary branch and a transient branch is rather conventional and commonly used in time series modeling. Likewise, the use of Kalman filtering is also standard. Therefore, the overall contribution of the paper appears to be relatively limited.

**Questions:**

please refer to the weakness

---

### Official Review · Reviewer_mZf2 · 2025-11-01

**Soundness:** 1
**Presentation:** 1
**Contribution:** 1
**Rating:** 2
**Confidence:** 4

**Summary:**

KIND proposes a two-branch forecasting model that separates stationary dynamics (a fixed Koopman/DMD-like operator) from transient dynamics (a transformer that emits time-varying operators), and then blends the two forecasts with a Kalman-inspired, uncertainty-weighted rule. The paper also introduces a staged curriculum to label windows as stationary vs transient, and reports results on ETTh2 (univariate oil temperature), TESLA cavity detuning, plus an appendix on M4 Hourly; KIND slightly beats Koopa and avoids Online-DMD divergence in open-loop forecasts. Claimed contributions are separation/blending of stationary & transient dynamics, learned uncertainty per branch, and a physics-informed pretraining/filtering pipeline.
What's most baffling in this paper is the lack of experimentation and literature review. The paper is severely lacking and needs major revision with thorough experimentation and related work discussion.

**Strengths:**

Clean factorization of global (stationary) vs local (transient) dynamics with a principled blend that mirrors inverse-variance fusion.

Uncertainty-aware design with heteroscedastic heads and a curriculum that teaches each branch “what it doesn’t know.”

**Weaknesses:**

* Core results use univariate ETTh2; no multivariate ETTh/ETTm, Weather, ECL, or Traffic, so it’s hard to assess generality. Baselines omit strong probabilistic forecasters (DeepAR, Deep State Space Models, TFT), which specifically target calibrated uncertainty in long-horizon forecasting.
* Comparisons within Koopman UQ. There’s no comparison to Bayesian/Probabilistic DMD or stochastic Koopman variants that already quantify epistemic/aleatoric uncertainty in operator learning.
* Severely lacking a detailed literature review :
   * Probabilistic/ stochastic Koopman & DMD: Bayesian DMD (IJCAI 2017; AAAI 2021 variants) and stochastic/ generator formulations (Klus et al.). These directly address uncertainty in operator learning.
   * DeepAR, Deep State Space Models, TFT (multi-horizon with interpretability). These are standard comparators for distributional accuracy & calibration.
   * Learned Kalman updates (close to their “Kalman-inspired” blend): KalmanNet and related neural Kalman gains.
   * Data assimilation & Koopman. Frion et al., Neural Koopman Prior for Data Assimilation (variational DA with Koopman prior). KODA (Koopman + DA; hybrid with explicit course-correction) – conceptually very close to KIND’s “stationary+residual” plus assimilation lens.

**Questions:**

See weaknesses.

---

### Official Review · Reviewer_PYoS · 2025-11-01

**Soundness:** 2
**Presentation:** 3
**Contribution:** 2
**Rating:** 2
**Confidence:** 4

**Summary:**

“KIND: Blending Stationary and Transient Koopman Dynamics with Learned Uncertainty” proposes KIND, a hybrid forecasting architecture that decomposes time series into stationary (model-driven) and transient (attention-driven) components, each governed by its own Koopman operator. Their outputs are fused through a Kalman-inspired, uncertainty-weighted blending mechanism, allowing the model to dynamically trust the more reliable branch at each time step. A staged training strategy teaches KIND to estimate and calibrate its own predictive uncertainty, enabling interpretable and adaptive forecasts. Experiments on superconducting RF cavity and electricity load datasets show that KIND outperforms both classical Dynamic Mode Decomposition (DMD) and neural baselines like Koopa, offering better accuracy, robustness to distributional shifts, and uncertainty awareness for time-varying systems

**Strengths:**

* **Originality:** While KIND builds on established ideas from Koopman operator theory and uncertainty modeling, its integration of stationary and transient dynamics via a Kalman-inspired fusion is a well-executed and meaningful refinement rather than a radical innovation. The framework thoughtfully extends prior Koopman and Transformer approaches to explicitly handle uncertainty and distributional shifts.
* **Quality:** The paper demonstrates technical rigor and careful experimentation. The staged training strategy, uncertainty calibration, and ablation studies are well designed, providing convincing evidence for the model’s effectiveness and stability. The comparisons with Online DMD and Koopa are fair and comprehensive.
* **Clarity:** The presentation is clear and systematic. The authors effectively explain complex components such as the uncertainty-weighted blending and lifting process, with helpful figures and equations. The paper maintains readability despite its technical depth.
* **Significance:** KIND makes a useful practical contribution by enhancing robustness and interpretability in time-series forecasting, particularly under nonstationary conditions. Although conceptually incremental, the model’s performance gains and uncertainty-aware design are valuable for applications in physics-informed and safety-critical forecasting.

**Weaknesses:**

* **Limited baseline coverage:** The primary weakness of the paper lies in its narrow experimental comparison. KIND is evaluated only against Koopa and Online DMD, both operator-based methods. This omits a wide range of relevant baselines—such as other dynamics-based models (e.g. KAE, Attraos, DeepEDM) and non-dynamics neural forecasters (e.g., Informer, Autoformer, DLinear). Including these would better contextualize KIND’s advantages and clarify whether its gains stem from the proposed blending mechanism or simply from model capacity.
* **Incomplete demonstration of generality:** The current evaluations focus mainly on two datasets, one physics-based and one benchmark. Broader comparisons across diverse domains would strengthen claims of adaptability and robustness.
* **Moderate novelty:** While well-executed, the conceptual leap, combining Koopman dynamics with Transformer-based transient modeling, is evolutionary rather than revolutionary, limiting its originality.
* **Empirical focus on interpretability without strong quantitative evidence:** The uncertainty calibration and interpretability claims are mostly qualitative; a more rigorous uncertainty evaluation (e.g., calibration plots) would substantiate these aspects.

Overall, while the approach is technically sound and promising, the lack of comprehensive baselines (similar to what is seen in TSF papers) substantially limits the paper’s empirical credibility and the strength of its comparative conclusions.

**Questions:**

1) More baseline comparisons are needed.
2) More comprehensive Dataset benchmarks are needed.
3) More detailed analysis of the uncertainty calibration would be great.
3) More discussion on dynamics based methods in the related work.

---

### Note · Authors · 2025-11-19

**Comment:**

The authors thank the reviewers for their time and constructive feedback. After careful consideration, we have decided to withdraw the submission in order to substantially revise the scope and experimental evaluation. We plan to resubmit an improved version in the future.

**Withdrawal Confirmation:**

I have read and agree with the venue's withdrawal policy on behalf of myself and my co-authors.